# Chunk-based Decoder for Neural Machine Translation

## Abstract

Chunks (or phrases) had once played a pivotal role in machine translation. By using a chunk rather than a word as the basic translation unit, local (intra-chunk) and global (inter-chunk) word orders/dependencies can be easily modeled. The chunk structure, despite its importance, has not been considered in the decoders of neural machine translation (NMT). In this paper, we propose chunk-based decoders for NMT, each of which consists of a chunk-level decoder and a word-level decoder. The chunk-level decoder models global dependencies, while the word-level decoder decides the word orders in a chunk. To generate a target sentence, the chunk-level decoder generates a chunk representation containing global information, based on which, the word-level decoder predicts the words inside the chunk. Experimental results show that our method can significantly improve translation performance in a WAT '16 English-to-Japanese translation task.

## 1 Introduction

Neural machine translation (NMT) performs an end-to-end translation based on a simple encoder-decoder model (Kalchbrenner and Blunsom, 2013; Sutskever et al., 2014; Cho et al., 2014b), and now has overwhelmed the classical, complex statistical machine translation (SMT) (Sennrich et al., 2016; Luong and Manning, 2016; Cromieres et al., 2016; Neubig, 2016). In NMT, an encoder first maps a source sequence into vector representations and a decoder then maps the vectors into a target sequence (§ 2). This simple framework allows researchers to incorporate the structure of the source

sentence as in SMT by leveraging various architectures as the encoder (Kalchbrenner and Blunsom, 2013; Sutskever et al., 2014; Cho et al., 2014b; Eriguchi et al., 2016b). Most of the NMT models, however, still rely on a sequential decoder based on recurrent neural network (RNN), due to the difficulty in capturing the structure of a target sentence that is unseen during translation.

With the sequential decoder, however, there are two problems to be solved. First, it is difficult to model long-distance dependencies (Bahdanau et al., 2015). A hidden state $h_t$ in an RNN is only conditioned by its previous output $y_{t-1}$, previous hidden state $h_{t-1}$ and current input $x_t$. This makes it difficult to capture the dependencies between an older output $y_{t-N}$ if they are too far from the current output. This problem can become more serious when the target sequence become longer. For example in Figure 1, when one translates the English sentence into the Japanese one, after the decoder predicts the content word "噛ま (bite)", it has to predict five function words "れ (*passive*)", "た (*past*)", "そう (*hearsay*)", "だ (*positive*)", and "けれど (*but*)" and a punctuation mark "、" before predicting the next content word "君 (you)". In such a case, the decoder is required to capture the longer dependencies in a target sentence.

Another problem with the sequential decoder is that it is expected to cover possible word orders simply by memorizing the local word sequences in the limited training data. This problem can be more serious in free word-order languages such as Czech, German, Japanese, and Turkish. In the case of the example in Figure 1, the order of the subject phrase "だれかが (someone was)" and the modifier phrase "犬に (by a dog)" are flexible. This means that simply memorizing the word order in training data is not enough to train a model that can assign a high probability to a correct sentence regardless of its word order.

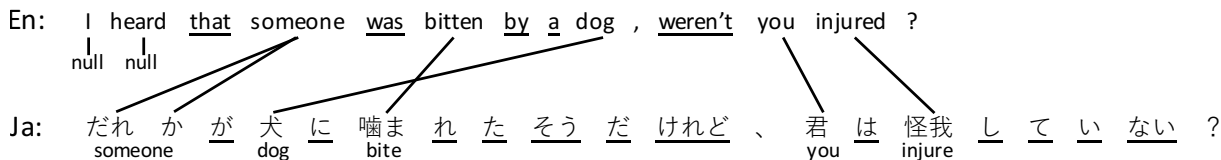

Figure 1: Translation from English to Japanese. The aligned words are content words and the underlined words are function words.

Looking back to the past, chunks (or phrases) are utilized to handle the above aforementioned problems in statistical machine translation (SMT) (Watanabe et al., 2003; Koehn et al., 2003) and in example-based machine translation (EBMT) (Kim et al., 2010). By using a chunk rather than a word as the basic translation unit, one can treat a sentence as a shorter sequence. This makes it easy to capture the longer dependencies in a target sentence. The order of words in a chunk is relatively fixed while that in a sentence is much more flexible. Thus, modeling intra-chunk (local) dependencies and inter-chunk (global) dependencies independently can help capture the difference of the flexibility between the word order and the chunk order in free word-order languages.

In this paper, we refine the original RNN decoder to consider chunk information in NMT. We propose three novel NMT models that capture and utilize the chunk structure in the target language (§ 3). Our focus is the hierarchical structure of a sentence: each sentence consists of chunks, and each chunk consists of words. To encourage an NMT model to capture the hierarchical structure, we start from a hierarchical RNN that consists of a chunk-level decoder and a word-level decoder (Model 1). Then, we improve the word-level decoder by introducing inter-chunk connections to capture the interaction between chunks (Model 2). Finally, we introduce a feedback mechanism to the chunk-level decoder to enhance the memory capacity of previous outputs (Model 3).

We evaluate the three models on the WAT '16 English-to-Japanese translation task (§ 4). The experimental results show that our best model outperforms the best single NMT model reported in WAT '16 (Eriguchi et al., 2016b).

Our contributions are twofold: (1) chunk information is firstly introduced into NMT to improve translation performance, and (2) a novel hierarchical decoder is devised to model the properties of chunk structure in the encoder-decoder framework.

## 2 Preliminaries: Attention-based Neural Machine Translation

In this section, we briefly introduce the architecture of the attention-based NMT model, which is the basis of our proposed models.

### 2.1 Neural Machine Translation

An NMT model usually consists of two connected neural networks: an encoder and a decoder. After the encoder maps a source sentence into a fixed-length vector, the decoder maps the vector into a target sentence. The implementation of the encoder can be a convolutional neural network (CNN) (Kalchbrenner and Blunsom, 2013), a long short-term memory (LSTM) (Sutskever et al., 2014; Luong and Manning, 2016), a gated recurrent unit (GRU) (Cho et al., 2014b; Bahdanau et al., 2015), or a Tree-LSTM (Eriguchi et al., 2016b). While various architectures are leveraged as an encoder to capture the structural information in the source language, most of the NMT models rely on a standard sequential network such as LSTM or GRU as its decoder.

Following (Bahdanau et al., 2015), we use GRU as the recurrent unit in this paper. A GRU unit computes its hidden state vector $h_i$ given an input vector $x_i$ and the previous hidden state $h_{i-1}$:

$$h_i = \text{GRU}(h_{i-1}, x_i). \quad (1)$$

The function $\text{GRU}(\cdot)$ is calculated as:

$$r_i = \sigma(\boldsymbol{W}_r x_i + \boldsymbol{U}_r h_{i-1} + \boldsymbol{b}_r), \quad (2)$$
$$z_i = \sigma(\boldsymbol{W}_z x_i + \boldsymbol{U}_z h_{i-1} + \boldsymbol{b}_z), \quad (3)$$
$$\tilde{h}_i = \tanh(\boldsymbol{W} x_i + \boldsymbol{U}(r_i \odot h_{i-1} + \boldsymbol{b})), \quad (4)$$
$$h_i = (1 - z_i) \odot \tilde{h}_i + z_i \odot h_{i-1}, \quad (5)$$

where the vectors $r_i$ and $z_i$ are reset gate and update gate, respectively. While the former gate allows the model to forget the previous states, the latter gate decides how much the model updates its content. All the $\boldsymbol{W}$s and $\boldsymbol{U}$s, or the $\boldsymbol{b}$s above

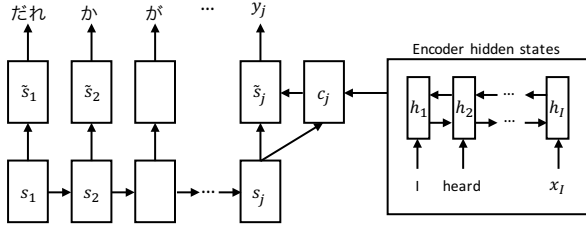

Figure 2: Standard word-based decoder.

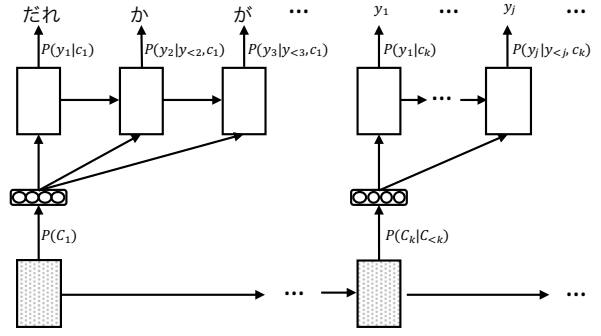

Figure 3: Chunk-based decoder. The top layer (word-level decoder) illustrates the first term in Eq. (15) and the bottom layer (chunk-level decoder) denotes the second term.

are trainable matrices or vectors. $\sigma(\cdot)$ and $\odot$ denote the sigmoid function and element-wise multiplication operator, respectively.

We train a GRU that encodes a source sentence $\{x_1, \cdots, x_I\}$ into a single vector $h_I$. At the same time, we jointly train another GRU that decodes $h_I$ to the target sentence $\{y_1, \cdots, y_J\}$. Here, the $j$-th word in the target sentence $y_j$ can be predicted with this decoder GRU and a nonlinear function $g(\cdot)$ followed by a softmax layer, as

$$c = h_I, \qquad (6)$$
$$s_j = \text{GRU}(s_{j-1}, [y_{j-1}; c]), \qquad (7)$$
$$\tilde{s}_j = g(y_{j-1}, s_j, c), \qquad (8)$$
$$P(y_j | \boldsymbol{y}_{<j}, \boldsymbol{x}) = \text{softmax}(\tilde{s}_j), \qquad (9)$$

where $c$ is a context vector of the encoded sentence and $s_j$ is a hidden state of the decoder GRU.

Following Bahdanau et al. (2015), we use a mini-batch stochastic gradient descent (SGD) algorithm with ADADELTA (Zeiler, 2012) to train the above two GRUs (i.e., the encoder and the decoder) jointly. The objective is to minimize the cross-entropy loss of the training data $\boldsymbol{D}$, as

$$J = \sum_{(\boldsymbol{x}, \boldsymbol{y}) \in \boldsymbol{D}} -\log P(\boldsymbol{y} | \boldsymbol{x}). \qquad (10)$$

### 2.2 Attention Mechanism for Neural Machine Translation

To use all the hidden states of the encoder and improve the translation performance of long sentences, Bahdanau et al. (2015) proposed using an attention mechanism. In their model, the context vector is not simply the last encoder state $h_I$ but rather the weighted sum of all encoder states, as follows:

$$c_j = \sum_{i=1}^{I} \alpha_{ji} h_i. \qquad (11)$$

Here, the weight $\alpha_{ji}$ decides how much a source word $x_i$ contributes to the target word $y_j$. $\alpha_{ji}$ is

computed by a feedforward layer and a softmax layer as

$$e_{ji} = v \cdot \tanh(\boldsymbol{W}_e h_i + \boldsymbol{U}_e s_j + \boldsymbol{b}_e), \qquad (12)$$
$$\alpha_{ji} = \frac{\exp(e_{ji})}{\sum_{j'=1}^{J} \exp(e_{j'i})}, \qquad (13)$$

where $\boldsymbol{W}_e$, $\boldsymbol{U}_e$ are trainable matrices and the $\boldsymbol{b}_e$ is a trainable vector. In a decoder using the attention mechanism, the obtained context vector $c_j$ in each timestep replaces $c$s in Eqs. (7) and (8). An illustration of the NMT model with the attention mechanism is shown in Figure 2.

The attention mechanism is expected to learn alignments between source and target words, and plays a similar role to the translation model in phrase-based SMT (Koehn et al., 2003).

## 3 Chunk-based Neural Machine Translation

Taking non-sequential information such as chunks (or phrases) structure into consideration is proved to be helpful for SMT (Watanabe et al., 2003; Koehn et al., 2003) and EBMT (Kim et al., 2010). We here focus on two important properties of chunks (Abney, 1991): (1) The word order in a chunk is almost always fixed; (2) A chunk consists of a few (typically one) content words surrounded by zero or more function words.

To fully utilize the above properties of a chunk, we propose to model the intra-chunk and the inter-chunk dependencies independently with a "chunk-by-chunk" decoder (See Figure 3). In the standard word-by-word decoder described in § 2, a target word $y_j$ in the target sentence $\boldsymbol{y}$ is predicted by taking the previous outputs $y_{<j}$ and the source

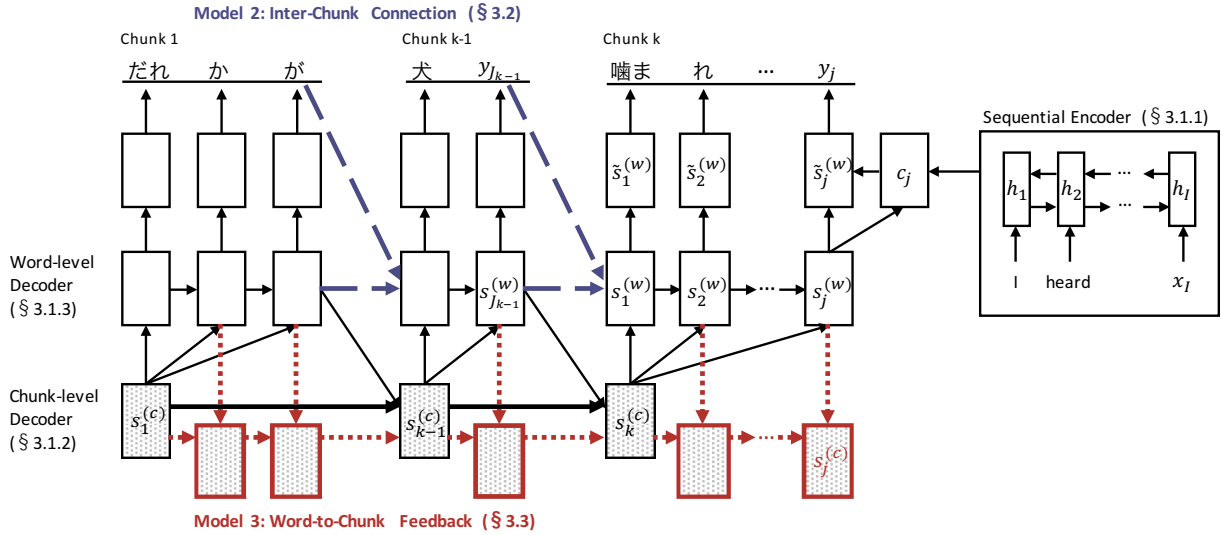

Figure 4: Proposed model: Chunk-based NMT. A chunk-level decoder generates a chunk representation for each chunk while a word-level decoder uses the representation to predict each word. The solid lines in the figure illustrate Model 1. The dashed blue arrows in the word-level decoder denote the connections added in Model 2. The dotted red arrows in the chunk-level decoder denote the feedback states added in Model 3; the connections in the thick black arrows are replaced with the dotted red arrows.

sentence $\boldsymbol{x}$ as input:

$$P(\boldsymbol{y}|\boldsymbol{x}) = \prod_{j=1}^{J} P(y_j|\boldsymbol{y}_{<j}, \boldsymbol{x}), \qquad (14)$$

where $J$ is the length of the target sentence. Not assuming any structural information of the target language, the sequential decoder has to memorize long dependencies in a sequence. To release the model from the strong pressure of memorizing the long dependencies over a sentence, we redefine this problem as the combination of a word prediction problem and a chunk generation problem:

$$P(\boldsymbol{y}|\boldsymbol{x}) = \prod_{k=1}^{K} \left\{ P(c_k|\boldsymbol{c}_{<k}, \boldsymbol{x}) \prod_{j=1}^{J_k} P(y_j|\boldsymbol{y}_{<j}, c_k, \boldsymbol{x}) \right\},$$
$$(15)$$

where $K$ is the number of chunks in the target sentence and $J_k$ is the length of the $k$-th chunk (see Figure 3). The first term represents the generation probability of a chunk and the second term indicates the probability of a word in the chunk . We model the former term as a chunk-level decoder and the latter term as a word-level decoder. As we will later confirm in § 4, both $K$ and $J_k$ are much shorter than the sentence length $J$, which is why our decoders do not have to memorize the long dependencies like the standard decoder does.

In the above formulation, we model the information of the words and their orders in a chunk.

No matter which language we target, a chunk usually consists of some content words and function words, and the word order in the chunk is almost always fixed (Abney, 1991). Although our idea can be used in several languages, the optimal network architecture could be adapted depending on the word order of the target language. In this work, we design models for languages in which content words are followed by function words, such as Japanese and Korean. The details of our models are described in the following sections.

### 3.1 Model 1: Standard Chunk-based NMT

The model described in this section is the basis of our proposed models. It consists of three parts: a sequential encoder (§ 3.1.1), a chunk-level decoder (§ 3.1.2), and a word-level decoder (§ 3.1.3). The part drawn in black solid lines in Figure 4 illustrates the architecture of Model 1.

#### 3.1.1 Sequential Encoder

We adopt a standard single-layer bidirectional GRU (Cho et al., 2014b; Bahdanau et al., 2015) as our encoder (see the right part in Figure 4). By using a standard sequential encoder, we need not perform any additional preprocessing on test data such as syntactic parsing. This design prevents our model from being affected by any errors that may occur as a result of additional preprocessing during test time.

### 3.1.2 Chunk-level Decoder

Our chunk-level decoder (see Figure 3) outputs a chunk representation. The chunk representation contains the information about the words that should be predicted by the word-level decoder.

To generate the representation of the $k$-th chunk $\tilde{s}_k^{(c)}$, the chunk-level decoder (see the bottom layer in Figure 4) takes the last states of the word-level decoder $s_{J_{k-1}}^{(w)}$ and updates its hidden state $s_k^{(c)}$ as:

$$s_k^{(c)} = \text{GRU}(s_{k-1}^{(c)}, s_{J_{k-1}}^{(w)}), \qquad (16)$$

$$\tilde{s}_k^{(c)} = \boldsymbol{W}_c s_k^{(c)} + \boldsymbol{b}_c. \qquad (17)$$

The obtained chunk representation $\tilde{s}_k^{(c)}$ continues to be fed into the word-level decoder until it outputs all the words in current chunk.

### 3.1.3 Word-level Decoder

Our word-level decoder (see Figure 4) differs from the standard sequential decoder described in § 2 in that it takes the chunk representation $\tilde{s}_k^{(c)}$ as input:

$$s_j^{(w)} = \text{GRU}(s_{j-1}^{(w)}, [\tilde{s}_k^{(c)}; y_{j-1}; c_{j-1}^{(w)}]), \qquad (18)$$

$$\tilde{s}_j^{(w)} = g(y_{j-1}, s_j^{(w)}, c_j^{(w)}), \qquad (19)$$

$$P(y_j|\boldsymbol{y}_{<j}, \boldsymbol{x}) = \text{softmax}(\tilde{s}_j^{(w)}). \qquad (20)$$

In a standard sequential decoder, the hidden state iterates over the length of a target sentence. In other words, its hidden layers are required to memorize the long-term dependencies in the target language. In contrast, in our word-level decoder, the hidden state iterates only over the length of a chunk. Thus, our word-level decoder is released from the pressure of memorizing the long (inter-chunk) dependencies and can focus on learning the short (intra-chunk) dependencies.

### 3.2 Model 2: Inter-Chunk Connection

The second term in Eq. (15) only iterates over a chunk ($j = 1$ to $J_k$). This means that the last state and the last output of a chunk are not being fed into the word-level decoder at the next timestep (see the black part in Figure 4). In other words, $s_1^{(w)}$ in Eq. (18) is always initialized before generating the first word in a chunk. This may affect the word-level decoder because it cannot access any previous information at the first word of each chunk.

To address this problem, we add new connections to Model 1 between the first state in a chunk

and the last state in the previous chunk, as

$$s_1^{(w)} = \text{GRU}(s_{J_{k-1}}^{(w)}, [\tilde{s}_k^{(c)}; y_{J_{k-1}}; c_{J_{k-1}}^{(w)}]). \qquad (21)$$

The dashed blue arrows in Figure 4 illustrate the added inter-chunk connections.

### 3.3 Model 3: Word-to-Chunk Feedback

The chunk-level decoder in Eq. (16) is only conditioned by $s_{J_{k-1}}^{(w)}$, the last word state in each chunk (see the black part in Figure 4). This may affect the chunk-level decoder because it cannot memorize what kind of information has already been generated by the word-level decoder. The information about the words in a chunk should not be included in the representation of the next chunk; otherwise, it may generate the same chunks for multiple times, or forget to translate some words in the source sentence.

To encourage the chunk-level decoder to remove the information about the previous outputs more carefully, we add feedback states to our chunk-level decoder in Model 2. The feedback state in the chunk-level decoder is updated at every timestep $j$ as:

$$s_j^{(c)} = \text{GRU}(s_{j-1}^{(c)}, s_j^{(w)}). \qquad (22)$$

The red lines in Figure 4 illustrate the added feedback states and their connection. The connections in the thick black arrows are replaced with the dotted red arrows in Model 3.

## 4 Experiments

### 4.1 Setup

**Data** To clarify the effectiveness of our decoders, we choose Japanese, a free word-order language, as the target language. Japanese sentences are easy to be broken into well-defined chunks (called *bunsetsus* (Hashimoto, 1934) in Japanese), and the accuracy of *bunsetsu*-chunking is over 99% (Murata et al., 2000; Yoshinaga and Kitsuregawa, 2014). The effect of chunking errors in training the decoder can be suppressed so we can evaluate the potential of our method. We use the English-Japanese training corpus in the Asian Scientific Paper Excerpt Corpus (ASPEC) (Nakazawa et al., 2016), which was provided in WAT '16. To remove inaccurate translation pairs, we extracted the first 2 million data from the 3 million translation pairs following the setting that gave the best performances in WAT '15 (Neubig et al., 2015).

| Corpus | # words | # chunks | # sentences |
|--------|---------|----------|-------------|
| Train | 44,286,317 | 13,707,397 | 1,505,871 |
| Dev. | 54,287 | - | 1,790 |
| Test | 54,088 | - | 1,812 |

Table 1: Statistics of the target language (Japanese) in extracted corpus after preprocessing.

**Preprocessings** For Japanese sentences, we performed tokenization using KyTea 0.4.7.[1] (Neubig et al., 2011) Then we performed *bunsetsu*-chunking with CaboCha 0.69.[2] For English sentences, we performed the same preprocessings described on the WAT '16 Website.[3] To suppress having possible chunking errors affect the translation quality, we removed extremely long chunks from the training data. Specifically, among the 2 million preprocessed translation pairs, we excluded the sentence pairs that matched any of following conditions: (1) The length of the source sentence or target sentence is larger than 64; (2) The maximum length of a chunk in the target sentence is larger than 8 (around 1% of whole data); (3) The maximum number of chunks in the target sentence is larger than 20 (around 2% of whole data). The amount of the excluded sentences by the condition (2) or (3) is negligible (less than 3% of whole data). Table 1 shows the details of the extracted data.

**Postprocessing** To perform unknown word replacement (Luong et al., 2015), we built a bilingual English-Japanese dictionary from all of the 3 million translation pairs. The dictionary was extracted with the MGIZA++ 0.7.0[4] (Och and Ney, 2003; Gao and Vogel, 2008) word alignment tool by automatically extracting the alignments between English words and Japanese words.

**Evaluation** Following the WAT '16 evaluation procedure, we used BLEU (Papineni et al., 2002) and RIBES (Isozaki et al., 2010) to evaluate our models. The BLEU scores were calculated with `multi-bleu.pl` in Moses 2.1.1[5] (Koehn et al., 2007); RIBES scores were calculated with `RIBES.py` 1.03.1.[6] (Isozaki et al., 2010)

---

[1]http://www.phontron.com/kytea/
[2]http://taku910.github.io/cabocha/
[3]http://lotus.kuee.kyoto-u.ac.jp/WAT/baseline/dataPreparationJE.html
[4]https://github.com/moses-smt/mgiza
[5]http://www.statmt.org/moses/
[6]http://www.kecl.ntt.co.jp/icl/lirg/ribes/index.html

| | |
|---|---|
| $\rho$ of ADADELTA | 0.95 |
| $\epsilon$ of ADADELTA | $1e^{-6}$ |
| Initial learning rate | 1.0 |
| Gradient clipping | 1.0 |
| Mini-batch size | 64 |
| Dimension of hidden states | 1024 |
| Dimension of embedding vectors | 1024 |

Table 2: Hyperparameters for training.

**Training Details** We use a single layer bidirectional GRU for the encoder and standard single layer GRUs for the word-level decoder and the chunk-level decoder. The vocabulary sizes are set to 30k for both source and target languages. The conditional probability of each target word is computed with a deep-output (Pascanu et al., 2014) layer with maxout (Goodfellow et al., 2013) units. The maximum number of output chunks is set to 20 and the maximum length of a chunk is set to 9.

The models are optimized using ADADELTA following (Bahdanau et al., 2015). The hyperparameters of the training procedure are fixed to the values given in Table 2. Note that the learning rate is halved when the BLEU score on the development set does not increase for 30,000 batches. All the parameters are initialized randomly with Gaussian distribution. It takes about a week to train each model with an NVIDIA TITAN X (Pascal) GPU.

## 4.2 Results

Following (Cho et al., 2014a), we perform beam search[8] with length-normalized log-probability to decode target sentences. We saved the trained models that performed best on the development set during training, and use them to test the systems with the test set. Table 3 shows the results on the test set. Note that all the models listed in Table 3, including our three models, are single models without ensemble techniques. We set the word-based sequence-to-sequence model (Li et al., 2016) as our baseline, which is a standard implementation of the attention-based NMT described in § 2. We also compare our methods with the tree-to-sequence model (Eriguchi et al., 2016b) to compare the effectiveness of capturing the structure in the source language and that in the target language. Our improved models (Model 2 and Model 3) outperform all the single models reported in WAT '16. The best model (Model 3)

---

[7]http://lotus.kuee.kyoto-u.ac.jp/WAT/evaluation
[8]Beam size is set to 20.

| System | RNN | $|V_{src}|$ | $|V_{trg}|$ | BLEU | RIBES |
|---|---|---|---|---|---|
| Word-based Seq-to-Seq  (Li et al., 2016) | GRU | 40k | 30k | 33.47 | 78.75 |
| Word-based Tree-to-Seq  (Eriguchi et al., 2016b) | LSTM | 88k | 66k | 34.87 | 81.58 |
| Character-based Tree-to-Seq (Eriguchi et al., 2016a) | LSTM | 88k | 3k | 31.52 | 79.39 |
| Proposed chunk-based model (Model 1) | GRU | 30k | 30k | 33.56 | 79.92 |
| + Inter-chunk connection (Model 2) | GRU | 30k | 30k | 35.44 | 80.95 |
| + Word-to-chunk feedback (Model 3) | GRU | 30k | 30k | **36.20** | **82.06** |

Table 3: The results and settings of the baseline systems and our systems. $|V_{src}|$ and $|V_{trg}|$ denote the vocabulary size of the source language and the target language, respectively. Only single NMT models (w/o ensembling) reported in WAT '16 are listed here. Full results are available on the WAT '16 Website.[7]

Source: this paper described the new author 's theory on glass transition point from crosslinked resin to linear polymer .

Reference: 著者 の 架橋 樹脂 から 線型 ポリマへの ガラス 転移点 に 関 する 新しい 理論 に ついて 述べた。

Model 1: 架橋 樹脂 から 線状 高 分子 に ガラス 転移 点 に 関 する 新しい 著者の 著者の 理論 に つい て 述べた。
from crosslinked resin   to linear polymer   about glass transition point   new   author's   author's   the theory   described

Model 2: 架橋 樹脂 から 線状 ポリマへの ガラス 転移 点 に 関 する 著者の 理論 に ついて 述べた。
from crosslinked resin   to linear polymer   about glass transition point   author's   the theory   described

Model 3: 架橋 樹脂 から 線状 ポリマーへの ガラス 転移 点 に 関 する 著者の 新しい 理論 に ついて 述べた。
from crosslinked resin   to linear polymer   about glass transition point   author's   new   the theory   described

Figure 5: Translation examples of a long sentence. Each sequence of underlined words correspond to a chunk recognized by our decoder. Model 1 outputs a chunk "著者の (author's)" twice by mistake. Model 2 does not output a chunk "新しい (new)" by mistake. In contrast, Model 3 outputs a correct translation although there is a minor word order difference from the reference.

outperforms the tree-to-sequence model (Eriguchi et al., 2016b) by +1.33 BLEU score and +0.48 RIBES score. The results show that capturing the chunk structure in the target language is more effective than capturing the syntax structure in the source language. Compared with the character-based NMT model, our Model 3 outperformed the model of (Eriguchi et al., 2016a) by +4.68 BLEU score and +2.67 RIBES score. The character-based model has a great advantage in that it does not require a large vocabulary size. Although the character-based model is less time-consuming thanks to the small target vocabulary size ($|V_{trg}| = 3k$), our chunk-based model significantly outperformed it in terms of translation quality. One possible reason for this is that using a character-based model rather than a word-based model makes it more difficult to capture long-distance dependencies because the length of a target sequence becomes much longer in the character-based model.

To understand the qualitative difference between our three models, we show translation examples in Figure 5. While Model 3 outputs a cor-rect translation, there are some errors in the outputs of Model 1 and Model 2. Only Model 1 outputs a chunk "著者の (author's)" twice continuously. This error indicates that the inter-chunk connections added in Model 2 play important roles in memorizing previous word states more efficiently. On the other hand, Model 2 does not output a chunk "新しい (new)" by mistake, which is probably because it does not have a good ability to memorize the previous chunks. This phenomenon supports the importance of the feedback states that are added in Model 3.

## 5 Related Work

There has been much work done on using chunk (or phrase) structure to improve machine translation quality. The most notable work was phrase-based SMT (Koehn et al., 2003), which has been the basis for a huge amount of work on SMT for more than ten years. Apart from this, Watanabe et al. (2003) proposed a chunk-based translation model that generates output sentences in a chunk-by-chunk manner. The chunk structure is effective

not only for SMT but also for example-based machine translation (EBMT). Kim et al. (2010) proposed a chunk-based EBMT and showed that using chunk structures can help with finding better word alignments. Our work is different from their works in that our models are based on NMT, but not SMT or EBMT. The decoders in the above works can model the chunk structure by storing chunk pairs in a large table. On the other hand, we do that by separately training a chunk generation model and a word prediction model with two RNNs.

While most of the NMT models focus on the conversion between sequential data, some works have tried to incorporate non-sequential information into NMT (Eriguchi et al., 2016b; Su et al., 2017). Eriguchi et al. (2016b) use tree-based LSTM (Tai et al., 2015) to encode input sentence into context vectors. Given a syntactic tree of a source sentence, their tree-based encoder encodes words from the leaf nodes to the root nodes recursively. Su et al. (2017) proposed a lattice-based encoder that considers multiple tokenization results while encoding the input sentence. To prevent the tokenization errors from propagating to the whole NMT system, their lattice-based encoder can utilize multiple tokenization results. These works focus on the encoding process and propose better encoders that can exploit the structures of the source language. In contrast, our work focuses on the decoding process to capture the structure of the target language. The encoders described above and our proposed decoders are complementary so they can be combined into a single network.

Considering that our Model 1 described in § 3.1 can be seen as a hierarchical RNN, our work is also related to previous studies that utilize multi-layer RNNs to capture hierarchical structures in data. Hierarchical RNNs are used not only in the field of machine translation (Luong and Manning, 2016) but also for various NLP tasks such as document modeling (Li et al., 2015; Lin et al., 2015), dialog generation (Serban et al., 2017), image captioning (Krause et al., 2016), and video captioning (Yu et al., 2016). In particular, Li et al. (2015) and Luong and Manning (2016) use hierarchical sequence-to-sequence models, but not for the purpose of learning syntactic structures of target sentences. Li et al. (2015) build hierarchical models at the sentence-word level to obtain better document representations. Luong and Manning (2016) build the word-character level to cope with the out-of-vocabulary problem. In contrast, we build a chunk-word level model to explicitly capture the syntactic structure based on chunk segmentation.

In addition, the architecture of Model 3 is also related to stacked RNN, which has shown to be effective to improve the translation quality (Luong et al., 2015; Sutskever et al., 2014). Although these architectures look similar to each other, there is a fundamental difference between the directions of the connection between two layers. A stacked RNN consists of multiple RNN layers that are connected from the input side to the output side at every timestep. In contrast, our Model 3 has a different connection at each timestep. Before it generates a chunk, there is a feed-forward connection from the chunk-level decoder to the word-level decoder. However, after generating a chunk representation, the connection is to be reversed to feedback the information from the word-level decoder to the chunk-level decoder. By switching the connections between two layers, our model can capture the chunk structure explicitly. This is the first work that proposes the decoders for NMT that can capture plausible linguistic structures like chunk.

# 6 Conclusion

In this paper, we propose chunk-based decoders for NMT. As the attention mechanism in NMT plays a similar role to the translation model in phrase-based SMT, our chunk-based decoders are intended to capture the notion of chunk in chunk-based (or phrase-based) SMT. We utilize the chunk structure to efficiently capture long-distance dependencies and cope with the problem of free word-order languages like Japanese. We design three models that have hierarchical RNN-like architectures, each of which consists of a word-level decoder and a chunk-level decoder. We performed experiments on the WAT '16 English-to-Japanese translation task and found that our best model outperforms all the single models that were reported in WAT '16 by $+4.68$ to $+1.33$ BLEU scores and by $+3.31$ to $+0.48$ RIBES scores.

In future work, we will apply our method to other target languages and evaluate the effectiveness on different languages such as Czech, German or Turkish. In addition, we plan to combine our decoder with other encoders that capture language structure, such as a Tree-LSTM (Eriguchi et al., 2016b), or an order-free encoder, such as a CNN (Kalchbrenner and Blunsom, 2013).

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
