# Peer review of "Chunk-based Decoder for Neural Machine Translation"

_ACL 2017 — decision unknown_

[Official Review · Reviewer 1 · rating 4 · confidence 5]
soundness 5 · originality 3 · clarity 5 · impact 3 · substance 4 · appropriateness 5 · meaningful comparison 5 · presentation format Oral Presentation

- Strengths:
The paper presents an interesting extension to attention-based neural MT
approaches, which leverages source-sentence chunking as additional piece of
information from the source sentence. The model is modified such that this
chunking information is used differently by two recurrent layers: while one
focuses in generating a chunk at a time, the other focuses on generating the
words within the chunk. This is interesting. I believe readers will enjoy
getting to know this approach and how it performs.
The paper is very clearly written, and alternative approaches are clearly
contrasted. The evaluation is well conducted, has a direct contrast with other
papers (and evaluation tables), and even though it could be strengthened (see
my comments below), it is convincing.

- Weaknesses:
As always, more could be done in the experiments section to strengthen the case
for chunk-based models. For example, Table 3 indicates good results for Model 2
and Model 3 compared to previous papers, but a careful reader will wonder
whether these improvements come from switching from LSTMs to GRUs. In other
words, it would be good to see the GRU tree-to-sequence result to verify that
the chunk-based approach is still best.

Another important aspect is the lack of ensembling results. The authors put a
lot of emphasis is claiming that this is the best single NMT model ever
published. While this is probably true, in the end the best WAT system for
Eng-Jap is at 38.20 (if I'm reading the table correctly) - it's an ensemble of
3. If the authors were able to report that their 3-way chunk-based ensemble
comes top of the table, then this paper could have a much stronger impact.

Finally, Table 3 would be more interesting if it included decoding times. The
authors mention briefly that the character-based model is less time-consuming
(presumably based on Eriguchi et al.'16), but no cite is provided, and no
numbers from chunk-based decoding are reported either. Is the chunk-based model
faster or slower than word-based? Similar? Who know... Adding a column to Table
3 with decoding times would give more value to the paper.

- General Discussion:
Overall I think the paper is interesting and worth publishing. I have minor
comments and suggestions to the authors about how to improve their presentation
(in my opinion, of course). 

* I think they should clearly state early on that the chunks are supplied
externally - in other words, that the model does not learn how to chunk. This
only became apparent to me when reading about CaboCha on page 6 - I don't think
it's mentioned earlier, and it is important.

* I don't see why the authors contrast against the char-based baseline so often
in the text (at least a couple of times they boast a +4.68 BLEU gain). I don't
think readers are bothered... Readers are interested in gains over the best
baseline.

* It would be good to add a bit more detail about the way UNKs are being
handled by the neural decoder, or at least add a citation to the
dictionary-based replacement strategy being used here.

* The sentence in line 212 ("We train a GRU that encodes a source sentence into
a single vector") is not strictly correct. The correct way would be to say that
you do a bidirectional encoder that encodes the source sentence into a set of
vectors... at least, that's what I see in Figure 2.

* The motivating example of lines 69-87 is a bit weird. Does "you" depend on
"bite"? Or does it depend on the source side? Because if it doesn't depend on
"bite", then the argument that this is a long-dependency problem doesn't really
apply.

[Official Review · Reviewer 2 · rating 4 · confidence 5]
soundness 5 · originality 3 · clarity 5 · impact 3 · substance 4 · appropriateness 5 · meaningful comparison 5 · presentation format Oral Presentation

- Summary

This paper introduces chunk-level architecture for existing NMT models. Three
models are proposed to model the correlation between word and chunk modelling
on the target side in the existing NMT models. 

- Strengths:

The paper is well-written and clear about the proposed models and its
contributions. 

The proposed models to incorporating chunk information into NMT models are
novel and well-motivated. I think such models can be generally applicable for
many other language pairs. 

- Weaknesses:

There are some minor points, listed as follows:

1) Figure 1: I am a bit surprised that the function words dominate the content
ones in a Japanese sentence. Sorry I may not understand Japanese. 

2) In all equations, sequences/vectors (like matrices) should be represented
as bold texts to distinguish from scalars, e.g., hi, xi, c, s, ...

3) Equation 12: s_j-1 instead of s_j.

4) Line 244: all encoder states should be referred to bidirectional RNN states.

5) Line 285: a bit confused about the phrase "non-sequential information such
as chunks". Is chunk still sequential information???

6) Equation 21: a bit confused, e.g, perhaps insert k into s1(w) like s1(w)(k)
to indicate the word in a chunk.  

7) Some questions for the experiments:

Table 1: source language statistics? 

For the baselines, why not running a baseline (without using any chunk
information) instead of using (Li et al., 2016) baseline (|V_src| is
different)? It would be easy to see the effect of chunk-based models. Did (Li
et al., 2016) and other baselines use the same pre-processing and
post-processing steps? Other baselines are not very comparable. After authors's
response, I still think that (Li et al., 2016) baseline can be a reference but
the baseline from the existing model should be shown. 

Figure 5: baseline result will be useful for comparison? chunks in the
translated examples are generated *automatically* by the model or manually by
the authors? Is it possible to compare the no. of chunks generated by the model
and by the bunsetsu-chunking toolkit? In that case, the chunk information for
Dev and Test in Table 1 will be required. BTW, the authors's response did not
address my point here. 

8) I am bit surprised about the beam size 20 used in the decoding process. I
suppose large beam size is likely to make the model prefer shorter generated
sentences. 

9) Past tenses should be used in the experiments, e.g.,

Line 558: We *use* (used) ...

Line 579-584: we *perform* (performed) ... *use* (used) ...

...

- General Discussion:

Overall, this is a solid work - the first one tackling the chunk-based NMT;
and it well deserves a slot at ACL.